# Overview of the Molecular Modalities and Signaling Pathways Intersecting with β-Amyloid and Tau Protein in Alzheimer's Disease

Ahmed M. Elshazly [1,2], Melanie M. Sinanian [2], Diaaeldin M. Elimam [3,4] and Sherin Zakaria [1,*]

[1] Department of Pharmacology and Toxicology, Faculty of Pharmacy, Kafrelsheikh University, Kafrelsheikh, P.O. Box 33516, Egypt; elshazlyam@vcu.edu
[2] Department of Pharmacology and Toxicology, Virginia Commonwealth University, Richmond, VA 23298, USA; melanie.sinanian@vcuhealth.org
[3] Department of Pharmacognosy, Faculty of Pharmacy, Kafrelsheikh University, Kafrelsheikh, P.O. Box 33516, Egypt; dr_deya@pharm.kfs.edu.eg
[4] School of Molecular and Cellular Biology, Faculty of Biological Sciences, University of Leeds, Leeds LS2 9JT, UK
[*] Correspondence: sherin_zakaria@pharm.kfs.edu.eg

**Abstract:** Alzheimer's disease (AD) is one of the major causes of dementia and its incidence represents approximately 60–70% of all dementia cases worldwide. Many theories have been proposed to describe the pathological events in AD, including deterioration in cognitive function, accumulation of β-amyloid, and tau protein hyperphosphorylation. Infection as well as various cellular molecules, such as apolipoprotein, micro-RNA, calcium, ghrelin receptor, and probiotics, are associated with the disruption of β-amyloid and tau protein hemostasis. This review gives an overview on the integrative cellular and signaling molecules that could play a complementary role in the dysregulation of β-amyloid and tau proteins.

**Keywords:** Alzheimer's; β-amyloid; tau protein; calcium; probiotics; apolipoprotein; infection

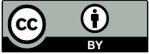

## 1. Introduction

Alzheimer's disease (AD) is the most frequent cause of dementia, accounting for 60–70% of all cases globally [1,2]. In 2019, AD was ranked the sixth most common cause of death in the US, while in 2020 and 2021 AD was the seventh leading cause of death in the US [3]. Furthermore, AD is the fifth most common cause of death in US citizens aged 65 and older [4]. Various molecular theories have been proposed to explain the pathological events in AD, including the accumulation of β-amyloid and hyperphosphorylation and aggregation of tau protein. The disruption of β-amyloid and tau protein hemostasis are associated with several cellular molecules and interactions, such as apolipoprotein, micro-RNA, calcium, and ghrelin receptor (GHSR1α). Gut microflora and infection are also associated with their dysregulation. This review will shed a light on the integrating cellular and signaling molecules that may have a complementary role in β-amyloid and tau protein dysregulation (Figure 1).

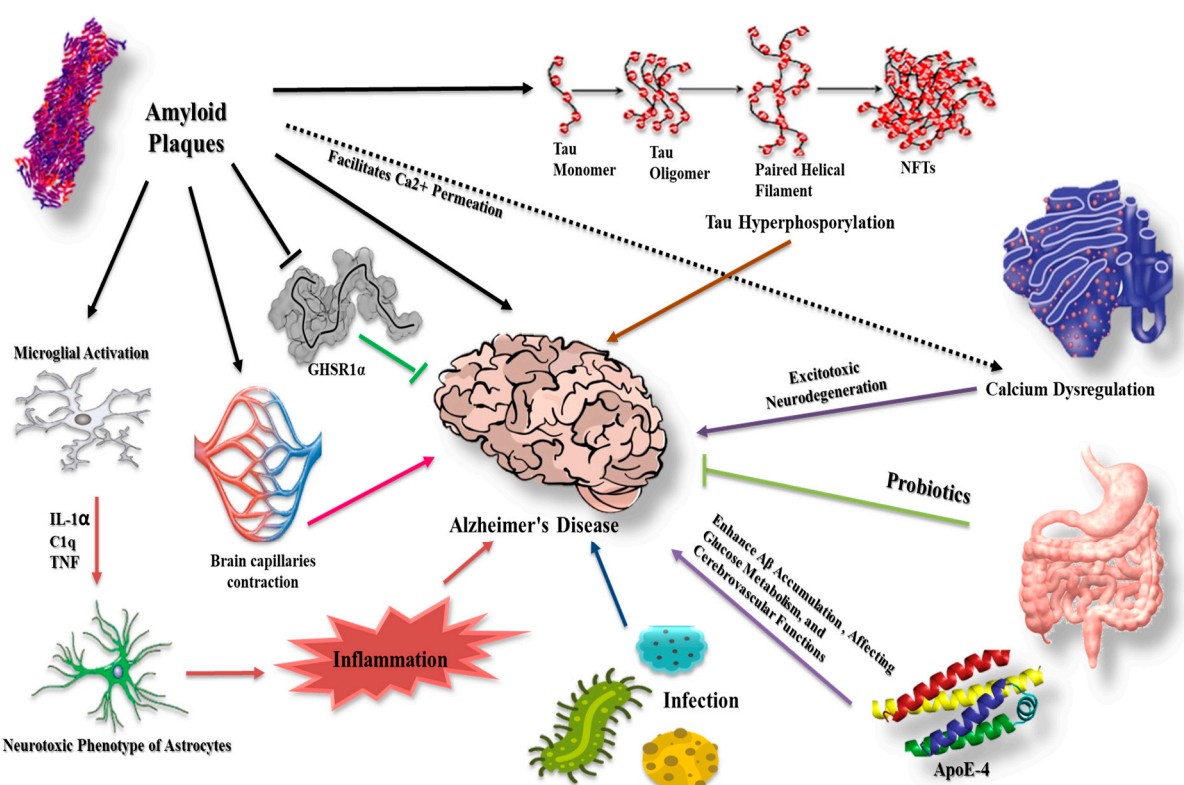

**Figure 1.** Different molecular modalities and signaling pathways intersecting with β-amyloid and tau protein in AD. β-amyloid plaques affect several pathways that may contribute to AD progression. For example, they facilitate Ca$^{2+}$permeation, eventually leading to neurotoxicity, and increase the extent of inflammation via microglial and astrocyte activation. Furthermore, β-amyloid counteracts the protective effect of GHSR1α and causes cerebral capillary vasocontraction while potentiating tau phosphorylation. Tau hyperphosphorylation has a role in AD etiology together with infection and apolipoprotein E-4 (ApoE-4). Probiotics demonstrate protective effects.

## 2. β-Amyloid in Alzheimer's Disease

β-amyloid (Aβ) aggregation has long been thought to be a key and first event in the etiology of Alzheimer's disease. Amyloid protein precursors (APPs) are enzymatically cleaved into Aβ peptides that range in length from 38 to 43 amino acids [5], giving rise to oligomers, polymers, and eventually insoluble amyloid aggregates upon linking with each other [6]. Aβ released by neurons, then enters the bloodstream and cerebrospinal fluid (CSF), with clearance mechanisms preventing Aβ deposition physiologically. However, the imbalance between Aβ production and clearance can result in its sedimentation [6], causing synaptic and neuronal dysregulation [7,8], and eventually contributing to AD pathogenesis [9,10]. Furthermore, the relationship between apoptosis and Aβ have been investigated, for example, Yao et al. [11] showed that Aβ affect the expression of Bcl2 family proteins. Aβ significantly reduce expression of the antiapoptotic proteins, *Bcl-w* and *Bcl-x*L, together with increasing the mitochondrial release of second mitochondrion-derived activator of caspase (Smac) [11]. Han et al. [12] reported that Aβ induces apoptosis via promoting mitochondrial fission, disrupting mitochondrial membrane potential, increasing intracellular reactive oxygen species (ROS) level, and activating the process of mitophagy. Barrantes et al. [13] also showed that Aβ 42 causes DNA strand breaks, leading to p53-mediated apoptosis.

## 3. Tau Protein and Formation of Neurofibrillary Tangles

Neurofibrillary pathology is one of the most common characteristics of AD, and includes neurotic plaques, neurofibrillary tangles, and threads generated by tau protein

aggregation, which is a microtubule-associated protein [6]. In pathological situations, tau undergoes hyperphosphorylation**,** forming insoluble fibers ("paired helical filaments") [14]. Hyperphosphorylated paired helical filaments combine to generate neurofibrillary tangles (NFTs).

Physiologically, tau phosphorylation is controlled by the balance between kinases and phosphatases [14,15]. This balance can be disrupted via oxidative stress and through an increase in the activity of protein kinases, mainly glycogen synthase kinase 3 (GSK-3$\beta$), which has been shown to be upregulated in AD [15]. Furthermore, the reduced phosphatase activity in the brain of AD patients can augment the hyperphosphorylation induced by protein kinases [16,17].

Lovell et al. [18] has shown that tau phosphorylation is significantly increased with higher GSK-3$\beta$ activity in primary rat cortical neuron cultures, stimulated through cuprizone, a copper chelator, in combination with oxidative stress (Fe$^{2+}$/H$_2$O$_2$) [18]. Moreover, they demonstrated a reduction in tau hyperphosphorylation through the inhibition of GSK-3$\beta$ activity with lithium, as confirmed by transglutaminase 3 staining [18]. Consistently with Lovell et al., Su et al. [19] showed that a fragment of tau protein possesses copper reduction activity, initiating the copper-mediated generation of hydrogen peroxide. The generated hydrogen peroxide has been shown to increase GSK-3$\beta$ activity, causing tau hyperphosphorylation in human embryonic renal cells 293 [17].

Tau hyperphosphorylation also affects the stability of microtubules, resulting in axonal and neural dysfunction [20]. Therefore, many efforts have been made to utilize microtubule-stabilizing drugs in AD. For example, Zhang et al. [21] showed that the administration of paclitaxel in mouse models with tau pathology restored fast axonal transport in spinal axons and improved motor impairment. Recently, Zhang et al. [22] showed that triazolopyrimidine, a microtubule-stabilizing drug, significantly lowered tau pathology and improved cognitive function in transgenic mouse models of tauopathy.

## 4. Prion-Like Conformation of β-Amyloid and Tau Proteins

Prion protein (PrPSc) is a unique protein form that has enhanced infectivity, self-replication ability, and persistent survival—even in the denaturing conditions of the gut. In humans, prions can cause different neuronal disorders, including Creutzfeldt–Jakob disease (CJD), and fatal familial insomnia (FFI) [23], which can either develop spontaneously from hereditary factors or as a result of infection.

A$\beta$ aggregates into different forms, including polymorphic amyloid fibrils and a variety of intermediate assemblies, including oligomers and protofibrils [24]. Several studies reported that A$\beta$ spreads through the brain in a harmful configuration similar to PrPSc [25]. Condello et al. demonstrated that the injection of brain-derived A$\beta$ from AD patients into the brains of transgenic mice exhibited a prion-like appearance [26] together with significant A$\beta$ deposition. In addition, in the human brain prion-like A$\beta$ forms in one or more regions before spreading to other regions, suggesting cross-synaptic transmission [27]. Pignataro et al. [28] discussed different findings to explain how A$\beta$ spreads through the brain regions. One hypothesis mentioned by Pignataro et al. [28] is that A$\beta$ can spread in the AD brain by advancing through synaptically connected regions since A$\beta$ is released from synaptic terminals, thus making brain nodes vulnerable to A$\beta$ accumulation.

Tau protein has been reported to spread in a prion-like manner in the brain, although earlier investigations focused mainly on total insoluble tau, as the presence of NFT correlates with the degree of brain atrophy and cognitive impairment in AD [29]. Clifford et al. reported that low activity of prion-like tau has been linked to extended life spans. Furthermore, they showed that both prion-like A$\beta$ and prion-like tau proteins were found in 100 postmortem brain tissue samples from patients who died of AD [30]. Levels of both prion-like A$\beta$ and prion-like tau are reported to be associated with age and dysregulation in production or clearance [27], however, more investigation is still re-

quired to uncover the mechanisms by which prion-like Aβ and prion-like tau proteins spread through the brain regions, in addition to the ability to target these conformations.

### 5. The Intervention of Glial Cells in β-Amyloid and Tau Protein Dysregulation

*5.1. Astrocytes*

Various neuronal cells, including astrocytes and microglia, help in maintaining the physiological levels of Aβ and tau protein. Astrocytes are specialized glial cells that construct the central nervous system (CNS) scaffold and can be found in two forms: fibrous (mostly in white matter) and protoplasmic (mostly in gray matter) [31]. Astrocytes contribute to various physiological activities, including fluid maintenance, cerebral blood flow regulation, neurotransmitter balance, and synaptic hemostasis [32]. Astrocytes also constitute the glymphatic system, which eliminates neurotoxic waste products such as amyloid and tau species [33]. Astrocytes, which are near amyloid plaques in human brains, have been found to contain amyloid-containing granules, implying that astrocytes work to remove amyloid accumulation during the disease progression [34]. In vitro and in vivo investigations have revealed that astrocytes move towards amyloid plaques and work to clear Aβ aggregation [35].

Astrocytes have two major phenotypes: A1, which is neurotoxic, and A2, which is thought to be protective. A1 astrocytes are expressed by activation of inflammatory cascades, mainly through NF-κB induction, a finding that is supported by the abundance of A1 astrocytes in postmortem brain tissues from persons with AD [36]. On the other hand, the A2 phenotype is induced by ischemia through activation of signal transducers and activators of the transcription 3 (STAT3) pathway. The neurotoxic A1 cells are marked by the expression of inflammatory mediators, while the protective A2 cells are marked by the expression of neurotrophic factors [37].

Interestingly, in animal models of AD, reactive astrocytes were discovered to release excessive GABA and glutamate, leading to impaired memory and synaptic loss [38]. Moreover, these cells contributed to microcirculation dysregulation and disruption of the blood–brain barrier (BBB), which facilitated Aβ accumulation and therefore disease progression [39]. Different molecules have been studied as possible treatments of AD through modulation of astrocyte phenotypes. One of these molecules is minocycline, where its intrathecal injection drastically downregulated A1 and upregulated A2 astrocyte levels [40,41].

*5.2. Microglia*

Microglia are innate immune cells of the myeloid lineage that exist in the CNS. Microglial activity is thought to be involved in CNS development, maturation, and senescence via the modulation of different regulatory networks [42]. Microglia play critical roles in neuronal apoptosis, synaptic maintenance, immune surveillance, and developmental synaptic pruning, in which the process of removing embryonic excess synapses improves the effectiveness of the neural network [42,43]. Dysregulated synaptic pruning is thought to be linked to autism disorders; moreover, it may be associated with weakened immune surveillance functions found in various neurodegenerative diseases [44,45].

Microglia express pattern recognition receptors (PRRs) recognizing two types of ligands: pathogen-associated molecular patterns (PAMPs) and damage-associated molecular patterns (DAMPs, including Aβ species). These receptors are responsible for triggering a microglial response in the presence of an exogenous or endogenous pathological insult [46]. The pathogenic species are then internalized by activated microglia via pinocytosis, phagocytosis, or receptor-mediated endocytosis. Through such endocytic processes, microglia attempt to degrade such pathogens and insults, additionally activating the release of various molecules, including interferons and chemokine receptors [47]. This process generally ceases once the immunological stimulus is removed, howev-

er, microglia have functional impairments and may be persistently activated in older brains, playing a role in AD pathogenesis [48].

Under pathological conditions, the morphology of microglia changes depending on the stage of the disease. In mouse models of AD, transcriptomic investigations have revealed that disease development is mirrored in microglia by progressive changes from a homeostatic to a disease-associated state, including less branching, which limits their surveillance functions [49]. The transition to the latter stage is often associated with the downregulation of homeostatic genes and upregulation of AD-associated genes, including apolipoprotein E (ApoE), protein tyrosine kinase binding protein (TYROBP), and triggering receptor expressed on myeloid cells 2 (TREM2) [50]. TREM2 plays a critical role in microglial activation processes, suggesting its possible role in the pathogenesis of neuronal disorders [51]. These studies confirmed the effects of aging on the human microglial phenotypes, including the downregulation of genes encoding cytoskeleton proteins, adhesion molecules, and cell surface receptors as well as the upregulation of certain genes, such as chemokine receptor type 4 (*CXCR4*), vascular endothelial growth factor 4 (*VEGF4*), and interleukin-15 (*IL-15*) [52].

### 5.3. Cellular Cross Talk between Astrocyte and Microglia

Aβ has been found to stimulate the NF-κB pathway in astrocytes, increasing complement C3 release, which then causes neuronal impairment and microglial activation [53]. The activated microglia secrete several factors, including interleukin 1 alpha (IL-1α), tumor necrosis factor (TNF), and complement component 1q (C1q), inducing astrocyte differentiation to the A1 phenotype [36]. Under AD inflammatory conditions, the interplay between astrocytes and microglia may generate a positive feedback loop, causing an inflammatory response [54].

## 6. Aβ Effect on the Cerebral Capillaries in Alzheimer's Disease

Angiogenesis abnormalities and decreased cerebral blood flow are considered to be the initial changes in early AD [55,56]. According to several studies, blood vessels in the brains of AD patients are connected inaccurately, causing a 42% reduction in gray matter blood flow. In animal experiments, exogenous Aβ has also been shown to diminish cerebral blood flow [27,57]. Nortley et al. demonstrated that Aβ accumulation caused vasoconstriction of brain capillaries by about 8.1% and reduced the energy supply in brains of AD patients, resulting in a reduction of the blood flow by 50%, which is approximate to the 42% drop that is reported in the gray matter of AD patients [58]. Aβ oligomer also participates significantly in the production of ROS, mostly by reduced nicotinamide adenine dinucleotide phosphate oxidase 4 (NOX4), and causes the release of endothelin (ET)-1, which subsequently binds to ETA receptors to cause pericyte contraction [58]. Pericytes become rigid and necrotic, producing persistent capillary constriction and ischemia. The vasoconstriction mechanism suggests that several prospective medications for the early treatment of Alzheimer's disease can prevent blood vessel contraction, including the vasodilator C-type natriuretic peptide (CNP) and the NOX4 inhibitor GKT137831 [58].

## 7. Aβ Reaction with Hippocampal Ghrelin/GHSR1α in Alzheimer's Disease

Growth hormone secretagogue receptor 1α (GHSR1α) (or ghrelin receptor) is a member of the G protein-coupled receptor (GPCR) family, which is known for its unique role in the hippocampus [59] by affecting eating-associated behaviors in the healthy hippocampus [60]. Furthermore, GHSR1α regulates the dopamine receptor D1 (DRD1), which mediates the activation of $Ca^{2+}$/calmodulin-dependent protein kinase II (CaMKII) through the noncanonical $G\alpha q$-$Ca^{2+}$ signaling pathway, which is important for hippocampus synaptic physiology and memory function [59,60]. Hippocampal lesions are one of the first lesions to occur in AD [61] and they have been linked to GHSR1α. The

dysregulation of GHSR1$\alpha$ has a significant impact on metabolic processes and calcium signaling of the hippocampus, both of which are associated with the deterioration of synaptic and memory functions in AD patients [62].

Interestingly, A$\beta$ has been shown to interact with GHSR1$\alpha$, inhibiting both GHSR1$\alpha$ activation and GHSR1a/DRD1 heterodimerization, causing hippocampus synaptic injury and memory impairments [63,64]. Thus, GHSR1$\alpha$ may represent a potential therapeutic target in AD. In animal and cell culture models, the GHSR1$\alpha$ agonists MK0677 and LY444711 demonstrated protective effects [65,66]; however, MK0677 failed to show clinical significance in AD patients. It has been shown that activating GHSR1$\alpha$ and DRD1 simultaneously with their selective agonists MK0677 and SKF81297, respectively, protects hippocampus synaptic and cognitive functions in AD mice models from A$\beta$-deleterious effects [63]. Moreover, the dual activation of GHSR1$\alpha$ and DRD1 stimulates neurogenesis in the dentate gyrus of AD mice models [62,63,67]. These data indicate the critical role of the GHSR1$\alpha$ signaling pathway in AD.

## 8. The Potential Role of Apolipoprotein in Alzheimer's Pathogenesis

Apolipoprotein E (ApoE) is a major cholesterol carrier that regulates lipid homeostasis by mediating lipid transportation from one tissue or cell type to another. In the periphery, ApoE is produced by the liver and macrophages. In the CNS, ApoE is produced mainly by astrocytes, transporting cholesterol to neurons via ApoE receptors, which are members of the low-density lipoprotein receptor (LDLR) family [68]. ApoE has been shown to influence several processes in the brain, including synaptic integrity, glucose metabolism, and cerebrovascular function [69]. ApoE has different isoforms, including ApoE-1, 2, 3, and 4, with ApoE-4 playing an important role in AD development [70].

Different mechanisms for A$\beta$ clearance are possible in the CNS, such as perivascular drainage and proteolytic degradation of A$\beta$ by proteases [71]. Among these mechanisms, ApoE isoforms play roles in A$\beta$ clearance from the brain interstitial fluid, which are mediated via ApoE-2 or ApoE-3. This is not the case with ApoE-4 [72,73]. In vivo studies revealed that both VLDLR and LRP1 cleared ApoE-2/A$\beta$ and ApoE-3/A$\beta$ complexes at the blood–brain barrier, while ApoE-4 bound to A$\beta$ altered the clearance pathway from LRP1 to the VLDL receptor (VLDLR) [74]. VLDLR promotes the internalization of the ApoE/A$\beta$ complex at a slower rate than LRP1, contributing to the delayed clearance of A$\beta$ [74]. ApoE-4 also influences other A$\beta$-degrading proteases, including neprilysin metalloprotease and insulin-degrading enzyme [71]. Miners et al. [75] and Cook et al. [76] have shown that AD patients who have at least one copy of ApoE-4 showed reduced expression of neprilysin in brain parenchyma and vasculature and downregulation in insulin-degrading enzyme level in the hippocampus [75,76]. One of the major A$\beta$ clearance mechanisms is the cellular absorption and subsequent breakdown of A$\beta$ by glial cells. Lin et al. recently showed that ApoE-4 homozygous astrocytes demonstrated reduced absorption of A$\beta$ 1–42 relative to ApoE-3 homozygous astrocytes in vitro. Furthermore, in human induced pluripotent stem cells (iPSC)-derived microglia-like cells, A$\beta$ clearance was observed to be reduced in the cells that expressed ApoE-4 compared to those that expressed ApoE-3 [73].

Interestingly, Christensen et al. [77] observed that ApoE-4 is highly expressed in postmortem brain tissues from individuals with AD. Furthermore, this protein is thought to enhance intraneuronal A$\beta$ accumulation [77], plaque deposition in the brain parenchyma [78], generation of A$\beta$ oligomers [79], and the severity of cerebral amyloid angiopathy [80]. In mouse models of AD disease, Rodriguez et al. [81] showed that ApoE-4 enhanced microglial reactivity towards A$\beta$ plaques, while Shi et al. [82] showed that ApoE-4 boosted proinflammatory activation and neurodegeneration of the microglia in tau-overexpressing transgenic mouse models [82]. Shi et al. [83] found that ApoE-4 contributes to changes in lipid structures on the microglial cell membranes, causing greater occurrences of the disease-associated microglia phenotype and deterioration of AD

pathogenesis and neurodegeneration, with these changes potentially being related to TREM2 [83].

Interestingly, the ApoE-4 genotype contributes to age-related decreases in brain glucose metabolism and affects insulin signaling independently of Aβ. Cerebral glucose hypometabolism is an early indicator of AD that can be found in presymptomatic people before disease onset [84]. Several epidemiological studies have reported that there are differences in insulin signaling in the brains of AD patients compared to normal controls [85]. Therefore, diabetes and midlife insulin resistance are potential risk factors for AD [86]. Furthermore, brain peroxisome proliferator-activated receptor (PPARγ) and PPAR coactivator 1 (PGC1α) signaling, which are important in the control of glucose metabolism and uptake, are downregulated in ApoE-4 gene-targeted replacement mice in comparison to ApoE-2 gene-targeted replacement mice [87]. In addition, ApoE-4 gene-targeted replacement mice exhibited more severe cognitive impairment, reduced cerebral blood volume, decreased glucose absorption, and altered insulin signaling after being fed a high-fat diet [88]. ApoE-4-targeted replacement mice also have changes in brain insulin signaling and insulin resistance [88]. Zhao et al. [88] showed that ApoE-4 binds to insulin receptors and traps them in endosomes, causing insulin receptor signaling to be disrupted [88].

These findings highlight the critical role of ApoE-4 in AD pathogenesis and its possibility to serve as a novel therapeutic target, which indeed requires further investigations.

## 9. Calcium as a Playmaker in Alzheimer's Disease

Recently, the association between calcium and Aβ has attracted huge attention. Recent studies showed that Aβ affects several types of voltage-gated calcium channels (VGCCs) via facilitating $Ca^{2+}$ passage through the plasma membrane, increasing postsynaptic calcium burden and eventually contributing to excitotoxic neurodegeneration [89,90]. Glutamate also plays a role, as $Ca^{2+}$ influx via VGCCs (N and P/Q types) directly triggers spontaneous glutamate release [91]. Furthermore, calcium entry at postsynaptic locations is facilitated by glutamate via L-type calcium channels and NMDA receptor channels. Aβ aids in the opening of the NMDA receptor, enabling an increase in the intracellular $Ca^{2+}$ concentration [89,90]. Memantine is an NMDA receptor antagonist drug that counteracts the effect of glutamate and consequently treats the AD symptoms [92].

### 9.1. Calcium Channels

One of the major reasons for disruption of $Ca^{2+}$ hemostasis is the excessive $Ca^{2+}$ influx through the L-type VGCC, which has been linked to AD etiology and thus makes these channels an effective target to help AD patients. Nimodipine, an L-type VGCC inhibitor, is being investigated for the treatment of senile dementia, including AD [93]. Another L-type VGCC inhibitor, nilvadipine, has been shown to lower Aβ levels and is currently being tested in a phase III clinical trial [94]. Furthermore, the L-type VGCC inhibitors, including isradipine, verapamil, diltiazem, and nifedipine, are suggested to have neuroprotective advantages [90,95], however, their efficacies in AD patients require further investigations.

ST101, a new cognitive enhancer that targets T-type VGCCs, has been shown to be beneficial for AD treatment [96]. In neuropathological injury and disease mouse models, ST101 enhanced synaptic plasticity, learning, and memory functions, most likely via increasing acetylcholine and dopamine release [96]. ST101 has also been reported to decrease Aβ levels in AD mice through inhibition of Aβ generation [97].

A plasma membrane $Ca^{2+}$ channel, $Ca^{2+}$ homeostasis modulator protein 1 (*CALHM1*), is significantly expressed in hippocampus neurons [98] and has been linked recently to AD etiology. Dreses-Werringloer et al. [99] showed that *CALHM1* expresses a multipass transmembrane glycoprotein that controls the cerebral cytosolic $Ca^{2+}$ concen-

trations and Aβ levels. Vingtdeux et al. [100] demonstrated the role of *CALHM1* in Aβ clearance, as *CALHM1* increases the extracellular secretion of Aβ, which is then degraded by insulin-degrading enzyme (IDE). *CALHM1* genetic variations have been associated with the onset of AD via disrupting $Ca^{2+}$ hemostasis and Aβ levels [98,101–103].

*9.2. Intracellular Calcium*

Emerging evidence suggests that the disruption of intracellular $Ca^{2+}$ homeostasis, particularly aberrant and excessive $Ca^{2+}$ release, plays a key role in AD neuropathology and has been associated with memory impairment and cognitive dysfunction [104].

$Ca^{2+}$ release from the endoplasmic reticulum (ER) has been shown to be increased in AD neurons [105] and several molecular mechanisms have been proposed to explain this excessive release. Among these mechanisms, presenilins (PS1 and PS2) has attracted attention as, upon alteration, they are recognized as deterministic genes that cause AD [106]. PS1 and PS2 are found in cellular membranes, including the ER [107], and comprise the catalytic core of the secretase complex. The secretase complex is responsible for cleaving APPs and therefore controlling Aβ levels [108]. PS mutations have been shown to disrupt the normal cleavage of APP and eventually Aβ production [108,109]. Furthermore, synaptic dysfunction in AD results from PS mutations, affecting $Ca^{2+}$ homeostasis. Cumulative evidence shows that the proteins encoded by mutated PS1 and PS2 interact with inositol triphosphate receptor (InsP3R), ryanodine receptor (RyR), sarco-endoplasmic reticulum calcium ATPase (SERCA) pump, ER and phospholipase. These interactions increase the overall sensitivity to calcium or the opening probability for calcium entry into the cell, thus increasing intracellular $Ca^{2+}$ concentration [110–114]. Pharmacological targeting of InsP3R and RyR using xestospongin and dantrolene, respectively, restored the $Ca^{2+}$ hemostasis and protected cells from apoptosis produced by Aβ [115,116]. RyR modulators, including RyCals and carvedilol, improved neuronal plasticity and synaptic transmission in AD mice models [117].

The sarcoplasmic reticulum has also been linked to Aβ formation via SERCA pump, which is critical for sequestering excessive cytoplasmic calcium. Emerging evidence has shown that SERCA activity is altered in AD. Inhibition of SERCA, either pharmacologically via thapsigargin or genetically, regulated $Ca^{2+}$ and reduced Aβ levels, suggesting that targeting SERCA could be a viable strategy in AD treatment [115].

## 10. The Role of MicroRNA-137 in the Onset and Progression of Alzheimer's Disease

MicroRNA-137 is an endogenous noncoding short molecule RNA that plays a role in the development and function of the nervous system. Interestingly, low levels of it have been associated with the onset and progression of AD [118]. Siegert et al.[119] showed that microRNA-137 inhibits extracellular Aβ sedimentation, controls calcium homeostasis, and regulates tau phosphorylation, suggesting its role in delaying the onset of AD. Serine palmitoyltransferase (SPT) is a key enzyme of sphingolipid metabolism and involved in ceramide and Aβ production, consisting of three subunits: serine palmitoyltransferase long chain 1 (SPTLC1), SPTLC2 and SPTLC3 [120–122]. SPTs contribute to Aβ production and eventually the death of neurons when levels are elevated via posttranscriptional regulation. MicroRNA-137 decreases Aβ production through posttranscriptional regulation of SPTLC1, giving it a protective effect [123,124]. Moreover, microRNA-137 corrects calcium dysregulation induced by CaV1.2 (L-type voltage-dependent calcium channel) overexpression [125]. Recently, Jiang et al. [126] showed that microRNA-137 inhibited the hyperphosphorylation of tau protein, with a possible effect on the L-type voltage-gated calcium channel subunit alpha-1C together with its associated gene CACNA1C **[126,127]**.

## 11. Gut Microbiota and Alzheimer's Disease

Recently, gut microbiota have attracted extensive attention because of their roles as critical regulators in various central processes, including immunological function, metabolic homeostasis, and neurological disorders [128]. The most studied probiotics belong to the genera *Lactobacillus* and *Bifidobacterium*, with some members that are natural occupants of the gut microbiota. In a recent clinical trial, Akbari et al. [129] showed that probiotic administration for 12 weeks causes a significant improvement in the cognitive function of AD patients, as assessed by the Mini-Mental State Examination (MMSE) [129].

Among the various fermentation products generated by the gut are short-chain fatty acids (SCFAs), including acetate, butyrate, lactate, and propionate [130]. In late-stage AD mouse models, an enteric bacterial metabolite, butyrate, has been found to block histone deacetylation (HDAc) and improve memory performance [131]. Nankova et al. [132] showed that butyrate and propionate have neuroprotective effects using PC12 cells, which have been widely employed as a general in vitro model to evaluate neuronal damage and neurotoxicity in AD [133]. In addition, they showed that SCFAs significantly lowered the expression of APPs [132]. Consistently with Nankova et al. [132], Kobayashi et al. [134] showed that the oral administration of *Bifidobacterium breve* strain A1 partially alleviated the cognitive decline of Aβ-induced AD mice [134]. Ho et al. [135] also showed that certain SCFAs effectively suppressed the production of detrimental Aβ aggregates [135].

In congruence with the protective effects of SCFAs, Smith et al. showed beneficial effects for SCFAs on immunity, as probiotic acetate supplementation demonstrated a reduction in neuroglia activation and proinflammatory cytokine expression in neuro-inflammatory rat models [136,137]. Moreover, SCFAs play a significant role in the maturation and activities of microglia [138]. It has also been reported that SCFAs modulate numerous signaling pathways, including NF-κB inhibition, HDAc inhibition, and activation of GPCRs [139]. In addition to signaling pathways, SCFAs cause modifications in cytokine production; alter the distribution and activity of natural killer cells, macrophages, granulocytes, and T cells; and promote mucosal and systemic antibody responses [137].

SCFAs have also been shown to control neurotransmitter production and neurotrophic genes, such as brain-derived neurotrophic factor (BDNF) and nerve growth factors [140,141]. BDNF signaling was found to be reduced in both the brain and the serum of AD patients [92]. This reduction in BDNF signaling was reversed by probiotic administration, as observed in rodent models [52,95,96]. These findings imply that SCFAs may modify crucial molecular signals, forming a network between gut microbiota and the host.

SCFAs are also well known for their significant anti-inflammatory effects [139]. Recently, probiotic utilization to target gut commensals was reported to reduce age-related inflammation and cognitive impairment [142]. The probiotic mixture SLAB51, consisting of *Streptococcus thermophilus* (DSM 32245), *B. lactis* (DSM 32246), *B. lactis* (DSM 32247), *L. acidophilus* (DSM 32241), *L. helveticus* (DSM 32242), *L. paracasei* (DSM 32243), *L. plantarum* (DSM 32244), and *L. brevis* (DSM 27961) [143], modified the microbial communities in 3xTg-AD mice, showing an increase in the proportions of *Bifidobacterium* spp. [144]. These findings imply that these bacteria may play a role in the regulation of inflammation in AD. The lower plasma concentrations of proinflammatory cytokines in AD animals treated with SLAB51 support this theory [144]. Furthermore, certain *Bifidobacterium* strains demonstrate anti-inflammatory capabilities through the inhibition of proinflammatory cytokine production by lipopolysaccharide-stimulated macrophages [145].

## 12. Infection and Alzheimer's Disease

Inflammation is confirmed to take place in AD brain tissue, as evidenced by the activation of microglia and complement system components [146]. Aβ has been reported to

act as an antimicrobial peptide that surrounds invaders in the brain and accumulate, forming plaques to protect it from further damage [147]. However, after APP mutations, the Aβ antimicrobial role is lost, increasing the extent of infection [27,148].

One of the suspected pathogenic factors in AD is the herpes virus. Herpes virus mediates Aβ accumulation and neuronal degeneration via suppressing miR-155 and HHV-6A. Additionally, herpes virus modifies several AD risk genes, including APBB2 and BACE1 [149]. Another pathogen, *Candida albicans*, which is responsible for oral ulcer infection, has been shown to cause a gelatinous granuloma, similar to AD, causing Aβ plaques and memory impairment [150]. The pathogen *Porphyromonas gingivalis* (*P. gingivalis*), which causes chronic periodontitis (CP), has also been linked to the formation of Aβ plaques. Upon infecting mice with *P. gingivalis*, the production of the harmful Aβ 1–42 increased with the activation of the complement pathway in the brain [148,151–153]. Moreover, the lipopolysaccharide of *P. gingivalis* has also been reported in the human AD brain, together with the high expression of gingipain, a *P. gingivalis* virulence factor [27,148,154]. *P. gingivalis* has been also linked to human ApoE [154], tau phosphorylation [27,155], and reduction in innate immune function. These data strongly suggest the possible role of *P. gingivalis* in the pathogenesis of AD, making *P. gingivalis* and gingipain possible effective targets in AD [27,154].

## 13. Conclusions

The pathogenic roles of Aβ and/or tau protein are associated with various complex events in brain tissues, including cellular interaction and molecular crosstalk. For example, astrocyte differentiation and microglial response to immunomodulatory molecules play important roles in the development of the pathogenic functioning of Aβ and/or tau protein. Other macro- and micromolecules, such as GHSR1α, ApoE-4, calcium, $H_2O_2$, and some microRNAmolecules, potentially modulated Aβ and tau expression and affected disease progression in experimental and clinical studies. Furthermore, infection and gut microbiota have been recently linked to Aβ and tau protein dysregulation.

Although there is controversy over whether Aβ and/or tau protein are among the major causes of AD pathogenesis [156–158], many drugs that target them are being investigated in various clinical trials, with lecanemab and aducanumab approved recently by the FDA (Table 1).

Furthermore, many studies have emerged recently discussing different hypotheses for Aβ and/or tau protein nucleation, for example, Kanaan et al. [159] showed that tau undergoes liquid–liquid phase separation and forms dynamic liquid droplets. These droplets serve as seeds for tau aggregation [160].

**Table 1.** Clinical trials involving β-amyloid and tau protein-targeted drugs.

| Drug | Action | References/Clinical Trials |
|---|---|---|
| ALZT-OP1a + ALZT-OP1b | Amyloid-related and antineuroinflammatory | NCT02547818, [161] |
| Plasma exchange with albumin 1 immunoglobulin | Removes amyloid | NCT01561053, [161] |
| AADvac-1 | Tau immunotherapy | [162,163], NCT02579252, NCT03174886 |
| ACI-35 | Tau immunotherapy | ISRCTN13033912, NCT04445831, [164] |
| ANAVEX2–73 | Anti-tau, anti-amyloid, and antineuroinflammatory | NCT03790709, [161] |
| GV-971 | Amyloid-related | NCT02293915, [161] |
| Crenezumab | Removes amyloid | NCT02670083, NCT03114657, NCT03491150, [161] |
| E2609 (elenbecestat) | Reduces amyloid production | NCT02956486, NCT03036280, [161] |
| BIIB076 | Tau immunotherapy | NCT03056729 |
| RG7345 | Tau immunotherapy | NCT02281786 |

| PNT001 | Tau immunotherapy | NCT04096287, NCT04677829 |
|---|---|---|
| Gantenerumab | Removes amyloid | NCT02294851, NCT02460094, NCT02658916, NCT03068468, NCT03352557, NCT03658135, NCT02051608, NCT01224106, NCT03444870, NCT03443973. |
| Solanezumab | Removes amyloid and prevent aggregation | NCT01760005, NCT02008357, [161] |
| Semorinemab | Extracellular tau | NCT02820896, NCT03289143, NCT03828747 |
| Bepranemab | Tau immunotherapy | NCT03464227, NCT03605082, NCT04185415, NCT04658199, NCT04867616 |
| JNJ-63733657 | Tau immunotherapy | NCT03375697, NCT03689153, NCT04619420. |
| Zagotenemab | Tau immunotherapy | NCT02754830, NCT03019536, NCT03518073 |
| Lu AF87908 | Tau immunotherapy | NCT04149860 |
| E2814 | Tau immunotherapy | NCT04231513 |
| Tilavonemab | Tau immunotherapy | NCT02494024, NCT03413319, NCT02985879, NCT03391765, NCT03744546, NCT02880956, NCT03712787. |
| Donanemab | Amyloid-related | NCT04437511 |
| Lecanemab | Blocks the formation of amyloid plaques in the brain | FDA-approved, NCT03887455 |
| Remternetug | Anti-amyloid, immunomodulator | NCT05463731 |
| Aducanumab | Anti-amyloid | FDA-approved, NCT02477800, [165] |

**Author Contributions:** All authors (A.M.E., M.M.S., D.M.E. and S.Z.) contributed to the development and writing. All authors have read and agreed to the published version of the manuscript.

**Funding:** This research received no external funding.

**Institutional Review Board Statement:** Not applicable.

**Informed Consent Statement:** Not applicable.

**Conflicts of Interest:** The authors declare no conflict of interest.

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
