# Peer review of "Overview of the Molecular Modalities and Signaling Pathways Intersecting with β-Amyloid and Tau Protein in Alzheimer’s Disease"

_2571-6980, doi:10.3390/neuroglia4030014_

Round 1

Reviewer 1 Report

This review brings together some important components of the pathogenesis of Alzheimer’s disease (AD).

Major weaknesses are:

1-      The logic in the general organization of the content is not apparent and should be more intentional to facilitate the transmission of information to readers for a better understanding of AD pathogenesis. For example, why not put the paragraph on Aβ and Tau aggregation (paragraph 5) just after the one about Aβ (paragraph 2) and the one about Tau (paragraph 3)? Why not put all the paragraphs about the cerebral components first, and then the 2 paragraphs about components influencing AD pathogenesis from the periphery (gut microbiota and infections) at the end? Please reorganize paragraphs in a more intentional way and briefly state in your introduction the logic behind the structure you choose.

2-      In paragraph 4 you wrongly classify microglia and astrocytes as neuronal cells, please change to glial cells.

3-      In paragraph 4.2 you wrongly classify DAMPs and PAMPs as surface receptors. DAMPs and PAMPs are the ligands of pattern recognition receptors (PRRs).

4-      In conclusion and Table 1, you talk about Iecanemab, please change to Lecanemab. Also, please add Aducanumab to your discussion and Table 1.

5-      In paragraph 2, you talk about accumulation of Aβ in the nucleus leading to apoptosis. Since this is not the main localization of Aβ, please briefly explain how it can end up in the nucleus. You also talk about Aβ being released from dying neurons, could you briefly explain what pools of neuronal Aβ are released? Is it coming from the cytosol, organelles, nucleus, directly from the plasma membrane? Then you mention the “2 hits hypothesis” but I did not understand the connection with what you said before or the reason for mentioning it here. Please explain more clearly.

6-      In paragraph 3, you talk about tau hyperphosphorylation as being controlled by the balance between kinases and phosphatases. Rather, there is a balance between the two enzymes in physiological situations, and tau hyperphosphorylation results from the unbalance between the two in pathological situations. Please rewrite to clarify this.

7-      Paragraph 5 is too vague, please clearly indicate the different aggregation forms of Aβ that have been linked to AD. It would be a good place to introduce the Aβ oligomers, since you mention them later in the review. You also mention that Aβ and tau aggregates spread throughout different regions of the brain suggesting cross-synaptic transmission. Please explain what this means, what is happening at the molecular level? Finally, you talk about more investigation being needed to uncover this area, please be more specific about what you mean. Maybe give an example or two of unresolved questions regarding the link between Aβ and tau conformation/aggregation and AD pathogenesis.

8-      In paragraph 9, you talk about the importance of calcium in AD and about the NMDA receptor. You could talk about memantine here to support this claim.

9-      The title of paragraph 9.1 is “extracellular calcium”, but I did not see where you talk about extracellular calcium in this paragraph. Please change the title or make the connection between the title and the content of the paragraph more apparent. In this paragraph you say that CALHM1 can control Aβ levels. Please briefly explain how.

10-   In paragraphs 9.2, 10, 11, and 12 almost all your statements are unclear and beg the question “how”? How are PS mutations affecting calcium homeostasis and expression of receptors? How are these receptors related to calcium? How is the Tg mouse model related to PS? How are ApoE, calcium, RyR, amyloid plaques, and tau related? Please rewrite paragraphs 9 to 12 to explicitly describe the connections or mechanisms you are mentioning.

 Minor weaknesses are:

1-      Please spell out GHSR1α the first time you use it. If possible, inserting a list of all the abbreviations would help.

2-      Paragraph 2, please change “…(APP) are broken into Aβ peptides…” by “…(APP) are enzymatically cleaved into Aβ peptides…”

3-      In paragraph 9 the sentence “Glutamate also plays…to an action potential.” does not make sense as written. How does a calcium channel release glutamate? Please rewrite it more clearly.

4-      In paragraph 9.1, the sentence “Moreover, ST101… cleaving APPS.” does not make sense as written. How do you decrease Aβ by cleaving APPs? Please rewrite it more clearly.

5-      Please make sure you correct minor grammar mistakes, missing or misspelled words, etc

The English language is generally good and understandable. Minor editing is required to correct a few grammatical errors and missing words.

Author Response

Reviewer 1:

This review brings together some important components of the pathogenesis of Alzheimer’s disease (AD).

Major weaknesses are:

1-      The logic in the general organization of the content is not apparent and should be more intentional to facilitate the transmission of information to readers for a better understanding of AD pathogenesis. For example, why not put the paragraph on Aβ and Tau aggregation (paragraph 5) just after the one about Aβ (paragraph 2) and the one about Tau (paragraph 3)? Why not put all the paragraphs about the cerebral components first, and then the 2 paragraphs about components influencing AD pathogenesis from the periphery (gut microbiota and infections) at the end? Please reorganize paragraphs in a more intentional way and briefly state in your introduction the logic behind the structure you choose.

A: Thank you for your comments. We adjusted the organization based on the reviewer comments.

2-      In paragraph 4 you wrongly classify microglia and astrocytes as neuronal cells, please change to glial cells.

A: Thank you for your comment. We corrected that to be as follow; “The intervention of glial cells in β-amyloid and tau proteins dysregulation”.

3-      In paragraph 4.2 you wrongly classify DAMPs and PAMPs as surface receptors. DAMPs and PAMPs are the ligands of pattern recognition receptors (PRRs).

A: Thank you for your comment. We corrected that to be as follow; “Microglia express a distinct pattern recognition receptors (PRRs). These receptors can recognize pathogenic species, cell debris, or dysregulated proteins (including Aβ species).”.

4-      In conclusion and Table 1, you talk about Iecanemab, please change to Lecanemab. Also, please add Aducanumab to your discussion and Table 1.

A: Thank you for your comment. We corrected that Iecanemab to Lecanemab and Aducanumab added.

5-      In paragraph 2, you talk about accumulation of Aβ in the nucleus leading to apoptosis. Since this is not the main localization of Aβ, please briefly explain how it can end up in the nucleus. You also talk about Aβ being released from dying neurons, could you briefly explain what pools of neuronal Aβ are released? Is it coming from the cytosol, organelles, nucleus, directly from the plasma membrane? Then you mention the “2 hits hypothesis” but I did not understand the connection with what you said before or the reason for mentioning it here. Please explain more clearly.

A: Thank you for your comment. We adjusted this section and removed the confusing information with more focus on the relationship between Beta-Amyloid and apoptosis to be as follow; The relations between apoptosis and Aβ have been investigated, for example, Yao et al. [10] showed that Aβ affect the expression of Bcl2 family proteins. Aβ significantly reduces expression of antiapoptotic Bcl-w and Bcl-xL, together with increasing the mitochondrial release of second mitochondrion-derived activator of caspase (Smac) [10]. Furthermore, Han et al. [11] reported that Aβ induce apoptosis via promoting mitochondrial fission, disrupting mitochondrial membrane potential, increasing intracellular ROS level as well as activating the process of mitophagy. Barrantes et al. [12] also showed that Aβ 42 cause DNA strand breaks, leading to p53-mediated apoptosis.

6-      In paragraph 3, you talk about tau hyperphosphorylation as being controlled by the balance between kinases and phosphatases. Rather, there is a balance between the two enzymes in physiological situations, and tau hyperphosphorylation results from the unbalance between the two in pathological situations. Please rewrite to clarify this.

A: Thank you for your comment. We rewrote this section in more clarified way.

7-      Paragraph 5 is too vague, please clearly indicate the different aggregation forms of Aβ that have been linked to AD. It would be a good place to introduce the Aβ oligomers, since you mention them later in the review. You also mention that Aβ and tau aggregates spread throughout different regions of the brain suggesting cross-synaptic transmission. Please explain what this means, what is happening at the molecular level? Finally, you talk about more investigation being needed to uncover this area, please be more specific about what you mean. Maybe give an example or two of unresolved questions regarding the link between Aβ and tau conformation/aggregation and AD pathogenesis.

A: Thank you for your comment. We adjusted this section according to the reviewer comments.

8-      In paragraph 9, you talk about the importance of calcium in AD and about the NMDA receptor. You could talk about memantine here to support this claim.

A: Thank you for your comment. We added Memantine as an example for drugs blocking glutamate effect and the section become as follow; Memantine, NMDA receptor antagonist, is an example for drugs that used to counteract the effect of glutamate and consequently treat the AD symptoms [87].

9-      The title of paragraph 9.1 is “extracellular calcium”, but I did not see where you talk about extracellular calcium in this paragraph. Please change the title or make the connection between the title and the content of the paragraph more apparent. In this paragraph you say that CALHM1 can control Aβ levels. Please briefly explain how.

A: Thank you for your comment. We adjusted the title of this section to be calcium channels. Furthermore, we clarify how CALHM1 control the amyloid levels as follow; Vingtdeux et al. [93] demonstrated the role of CALHM1 in Aβ clearance, as CALHM1 increase the extracellular secretion of Aβ, which then degraded by insulin-degrading enzyme (IDE).

10-   In paragraphs 9.2, 10, 11, and 12 almost all your statements are unclear and beg the question “how”? How are PS mutations affecting calcium homeostasis and expression of receptors? How are these receptors related to calcium? How is the Tg mouse model related to PS? How are ApoE, calcium, RyR, amyloid plaques, and tau related? Please rewrite paragraphs 9 to 12 to explicitly describe the connections or mechanisms you are mentioning.

A: Thank you for your comment. We adjusted the section “intracellular calcium” according to the reviewer comments. However, we think that the sections 10, 11, and 12 are clear and that’s supported by the other reviewers’ comments, who indicated minor comments in these parts.

Minor weaknesses are:

1-      Please spell out GHSR1α the first time you use it. If possible, inserting a list of all the abbreviations would help.

A: Thank you for your comment. We added the abbreviation as requested; Growth hormone secretagogue receptor 1α (GHSR1α) (or ghrelin re-ceptor) is a member of G protein-coupled receptor (GPCR) family…

2-      Paragraph 2, please change “…(APP) are broken into Aβ peptides…” by “…(APP) are enzymatically cleaved into Aβ peptides…”

A: Thank you for your comment. We substituted the statements as requested.

3-      In paragraph 9 the sentence “Glutamate also plays…to an action potential.” does not make sense as written. How does a calcium channel release glutamate? Please rewrite it more clearly.

A: Thank you for your comment. We adjusted the section to be as follow; Glutamate also plays a role, as Ca2+ influx via VGCCs (N and P/Q types) directly triggers spontaneous glutamate release [87].

4-      In paragraph 9.1, the sentence “Moreover, ST101… cleaving APPS.” does not make sense as written. How do you decrease Aβ by cleaving APPs? Please rewrite it more clearly.

A: Thank you for your comment. We adjusted the section to be as follow; Moreover, ST101 has also been reported to decrease Aβ level in AD mice via inhibiting Aβ generation [92].

5-      Please make sure you correct minor grammar mistakes, missing or misspelled words, etc

A: Thank you for your comment. We checked the manuscript for mistakes.

Reviewer 2 Report

The review entitled "Overview of the Molecular Modalities and Signaling Pathways Intersecting with β-Amyloid and Tau Protein in Alzheimer's Disease" by Elshazly et. al. described the overview of different pathways related to Amyloid and tau protein aggregation in AD. This review is well described piece of work with lots of information about AD related research. My comments are listed below-  1. Try to add most recent articles and information regarding AD and related various pathway. AD is currently the 7th cause of death in USA. 2. In tau pathology, microtubule disruption also plays role in AD. Briefly describe the role of microtubule disruption and microtubule targeted drug in tau pathology may be helpful for understanding. 3. Different markers or proteins changed in different signaling pathways, can the authors include those in the figure 1, which will be a good summary of AD related description. 4. What are the author's thoughts on phase separation of tau protein in AD  Thanks

Author Response

Reviewer 2:

The review entitled "Overview of the Molecular Modalities and Signaling Pathways Intersecting with β-Amyloid and Tau Protein in Alzheimer's Disease" by Elshazly et. al. described the overview of different pathways related to Amyloid and tau protein aggregation in AD. This review is well described piece of work with lots of information about AD related research. My comments are listed below- 

  1. Try to add most recent articles and information regarding AD and related various pathway. AD is currently the 7th cause of death in USA.

A: Thank you for your comment. We update our introduction as follow; Alzheimer's disease (AD) is the most frequent cause of dementia, ac-counting for 60–70% of all cases globally [1, 2]. In 2019, AD was ranked as the sixth most common cause of death in the US, while in 2020 and 2021; AD was the seventh-leading cause of death in the US [3]. Furthermore, AD is the fifth common cause of death in the US-citizens aged 65 and older [4].

  1. In tau pathology, microtubule disruption also plays role in AD. Briefly describe the role of microtubule disruption and microtubule targeted drug in tau pathology may be helpful for understanding.

A: Thank you for your comment. We update tau protein section according to the reviewer comments as follow; Tau hyperphosphorylation also affect the stability of microtubules, re-sulting in axonal and neural dysfunction [20]. Therefore, many efforts emerged to utilize microtubules-stabilizing drugs in AD. For example, Zhnag et al. [21] showed that the administration of paclitaxel for mouse models with tau pathology restored fast axonal transport in spinal axons and improved motor impairment. Recently, Zhang et al [22] showed that tria-zolopyrimidine, a microtubules-stabilizing drug, significantly lowered tau pathology and improved cognitive function in transgenic mouse models of tauopathy.

  1. Different markers or proteins changed in different signaling pathways, can the authors include those in the figure 1, which will be a good summary of AD related description.

A: Thank you for your comment. The goal of the figure is to give a simple overview of the manuscript content. And we think if we include more information, the figure will be difficult for the readers.

  1. What are the author's thoughts on phase separation of tau protein in AD  Thanks

A: Thank you for your comment. We added this section in the discussion as follow; Furthermore, many studies emerged discussing different hypotheses for β-Amyloid and/or Tau proteins nucleation, for example, Kanaan et al. [161] showed that tau undergoes liquid-liquid phase separation, forms dynamic liquid droplets. These droplets serve as seeds for tau aggregation [162].

Reviewer 3 Report

The authors have provided a comprehensive review encompassing various aspects of Alzheimer's disease, including its etiology, molecular mechanisms, and therapeutic trials. They begin by discussing the β-amyloid (Aβ) aggregation hypothesis, wherein amyloid protein precursors (APP) undergo breakdown, resulting in the formation of Aβ peptides that can form oligomers, polymers, and insoluble amyloid aggregates. While neurons release Aβ, its entry into the bloodstream and cerebrospinal fluid is typically countered by clearance mechanisms that prevent its accumulation.

Furthermore, the authors explore tau pathology and the involvement of kinases, such as GSK-3β, in the hyperphosphorylation of tau. They delve into the roles of the two major astrocyte phenotypes, A1 and A2, and their respective contributions to Alzheimer's disease. Microglial cells are also discussed in relation to their involvement in AD pathology, including the transition from a healthy to a diseased state, with a focus on the roles of ApoE, TYROBP, and TREM-2.

The review comprehensively covers the prion-like propagation of Aβ and tau in Alzheimer's disease. Additionally, the authors provide in-depth insights into the molecular aspects of the disease, including the interaction between GHSR1 and Aβ, resulting in the inhibition of GHSR1 activity. They elaborate on the effects of Aβ on cerebral capillaries and the significant role of Apolipoprotein in Alzheimer's disease, supported by recent references.

The review further explores the role of calcium signaling in Alzheimer's pathology, distinguishing between intracellular and extracellular calcium and their relevance to the disease. The emerging field of gut microbiota and its connection to Alzheimer's disease is summarized effectively, incorporating recent research developments. The relationship between infection and Alzheimer's disease, an area yet to be fully explored, is also addressed, with discussions on various pathogens studied in relation to the disease.

Moreover, the authors discuss the role of MicroRNA-137 in the onset and progression of Alzheimer's disease, shedding light on the underlying molecular mechanisms. Lastly, the review concludes with a list of diverse clinical trials conducted in Alzheimer's disease, accompanied by relevant references.

Overall, this review provides a comprehensive and engaging exploration of various aspects of Alzheimer's disease.

There are a few minor comments and suggestions for improvement before accepting the publication:

1.     In the section "5. Prion-like propagation of β-amyloid and Tau proteins," the third line of the second paragraph mentions "synthetic Aβ from AD patients," which doesn't make sense. It is understood that the authors want to discuss the prion-like activity of synthetic Aβ. Please rephrase the sentence to clarify this point.

2.     In the section "The potential role of apolipoprotein in Alzheimer's pathogenesis," the last line of the first paragraph states, "ApoE has different isoforms including ApoE-1, 2, 3, and 4, with ApoE4 playing an important role in AD development." The reference for this statement is missing.

3.     In the section "9.1 Extracellular Calcium," the seventh line of the first paragraph mentions, "Furthermore, L-type VGCC inhibitors, including isradipine, verapamil, diltiazem, and nifedipine, are suggested to have neuroprotective advantages." The reference for this statement is missing. Additionally, in the second line of the second paragraph, it is stated, "ST101, a new cognitive enhancer that targets T-type VGCC, has been shown to be beneficial for AD patients." The reference for this statement is missing as well.

4.     In the section "10. Gut microbiota and Alzheimer's disease," in the fifth paragraph, please provide more details about the composition of the SLAB51 probiotic mixture to enhance the explanation.

Overall, the article covers various aspects of Alzheimer's disease comprehensively and provides valuable insights.

Author Response

Reviewer 3:

The authors have provided a comprehensive review encompassing various aspects of Alzheimer's disease, including its etiology, molecular mechanisms, and therapeutic trials. They begin by discussing the β-amyloid (Aβ) aggregation hypothesis, wherein amyloid protein precursors (APP) undergo breakdown, resulting in the formation of Aβ peptides that can form oligomers, polymers, and insoluble amyloid aggregates. While neurons release Aβ, its entry into the bloodstream and cerebrospinal fluid is typically countered by clearance mechanisms that prevent its accumulation.

Furthermore, the authors explore tau pathology and the involvement of kinases, such as GSK-3β, in the hyperphosphorylation of tau. They delve into the roles of the two major astrocyte phenotypes, A1 and A2, and their respective contributions to Alzheimer's disease. Microglial cells are also discussed in relation to their involvement in AD pathology, including the transition from a healthy to a diseased state, with a focus on the roles of ApoE, TYROBP, and TREM-2.

The review comprehensively covers the prion-like propagation of Aβ and tau in Alzheimer's disease. Additionally, the authors provide in-depth insights into the molecular aspects of the disease, including the interaction between GHSR1 and Aβ, resulting in the inhibition of GHSR1 activity. They elaborate on the effects of Aβ on cerebral capillaries and the significant role of Apolipoprotein in Alzheimer's disease, supported by recent references.

The review further explores the role of calcium signaling in Alzheimer's pathology, distinguishing between intracellular and extracellular calcium and their relevance to the disease. The emerging field of gut microbiota and its connection to Alzheimer's disease is summarized effectively, incorporating recent research developments. The relationship between infection and Alzheimer's disease, an area yet to be fully explored, is also addressed, with discussions on various pathogens studied in relation to the disease.

Moreover, the authors discuss the role of MicroRNA-137 in the onset and progression of Alzheimer's disease, shedding light on the underlying molecular mechanisms. Lastly, the review concludes with a list of diverse clinical trials conducted in Alzheimer's disease, accompanied by relevant references.

Overall, this review provides a comprehensive and engaging exploration of various aspects of Alzheimer's disease.

There are a few minor comments and suggestions for improvement before accepting the publication:

  1. In the section "5. Prion-like propagation of β-amyloid and Tau proteins," the third line of the second paragraph mentions "synthetic Aβ from AD patients," which doesn't make sense. It is understood that the authors want to discuss the prion-like activity of synthetic Aβ. Please rephrase the sentence to clarify this point.

A: Thank you for your comment. We corrected the statement to be as follow;Condello et al. demonstrated that the injection of brain-derived Aβ from AD patients into the brains of transgenic mice, showed prion-like appearance”.

  1. In the section "The potential role of apolipoprotein in Alzheimer's pathogenesis," the last line of the first paragraph states, "ApoE has different isoforms including ApoE-1, 2, 3, and 4, with ApoE4 playing an important role in AD development." The reference for this statement is missing.

A: Thank you for your comment. We added the reference.

  1. In the section "9.1 Extracellular Calcium," the seventh line of the first paragraph mentions, "Furthermore, L-type VGCC inhibitors, including isradipine, verapamil, diltiazem, and nifedipine, are suggested to have neuroprotective advantages." The reference for this statement is missing. Additionally, in the second line of the second paragraph, it is stated, "ST101, a new cognitive enhancer that targets T-type VGCC, has been shown to be beneficial for AD patients." The reference for this statement is missing as well.

A: Thank you for your comment. We added the needed references.

  1. In the section "10. Gut microbiota and Alzheimer's disease," in the fifth paragraph, please provide more details about the composition of the SLAB51 probiotic mixture to enhance the explanation.

A: Thank you for your comment. We provide more details about SLAB51 composition as follow; The probiotic mixture SLAB51, consisting of Streptococcus thermophilus (DSM 32245), B. lactis (DSM 32246), B. lactis (DSM 32247), L. acidophilus (DSM 32241), L. helveticus (DSM 32242), L. paracasei (DSM 32243), L. plantarum (DSM 32244), and L. brevis (DSM 27961) [142]………

Overall, the article covers various aspects of Alzheimer's disease comprehensively and provides valuable insights.

Round 2

Reviewer 1 Report

Most of the revisions are satisfactory. However a few issues remain.

Abstract: GHSR1alpha appears for the first time, please either spell it out here or call it the ghrelin receptor.

5.2. Microglia, second paragraph: I would like to suggest writing: "Microglia express pattern recognition receptors (PRRs) recognizing two types of ligands; pathogen-associated molecular patterns (PAMPs) and damage-associated molecular patterns (DAMPs, including Abeta species)."

9., last 3 lines: I would like to suggest writing: "Memantine is an NMDA receptor antagonist drug that counteracts the effect of glutamate and consequently treats the AD symptoms."

9.2., second paragraph: "Cumulative evidence... the opening probability, ..." Please indicate the opening probability of what.

I still found a few typos and English issues.

4. Prion like conformation of ... (not confirmation)

5.3, line 1: NF-kB (not NF- kB)

6., line 5: GHSR1alpha (not GHSR1a)

9.1., last paragraph: "Vingtdeux..., which is then..."

Author Response

Abstract: GHSR1alpha appears for the first time, please either spell it out here or call it the ghrelin receptor.

A: Thank you very much for your comment. We add ghrelin receptor in the abstract instead of the abbreviation.

5.2. Microglia, second paragraph: I would like to suggest writing: "Microglia express pattern recognition receptors (PRRs) recognizing two types of ligands; pathogen-associated molecular patterns (PAMPs) and damage-associated molecular patterns (DAMPs, including Abeta species)."

A: Thank you very much for your comment. We adjusted this section to be as follow; Microglia express pattern recognition receptors (PRRs) recognizing two types of ligands; pathogen-associated molecular patterns (PAMPs) and damage-associated molecular patterns (DAMPs, including Aβ species). These receptors are responsible for triggering a microglial response in the presence of an exogenous or endogenous pathological insult [46]. 

9., last 3 lines: I would like to suggest writing: "Memantine is an NMDA receptor antagonist drug that counteracts the effect of glutamate and consequently treats the AD symptoms."

A: Thank you very much for your comment. We adjusted this section based on the reviewer comments. 

9.2., second paragraph: "Cumulative evidence... the opening probability, ..." Please indicate the opening probability of what.

A: Thank you very much for your comment. We adjusted this section to be as follow; the opening probability of calcium entry.

4. Prion like conformation of ... (not confirmation)

5.3, line 1: NF-kB (not NF- kB)

6., line 5: GHSR1alpha (not GHSR1a)

9.1., last paragraph: "Vingtdeux..., which is then..."

A: Thank you very much for your appreciated comments. We corrected these mistakes.